



# Characteristics and robustness of Agulhas leakage estimates: an inter-comparison study of Lagrangian methods

Christina Schmidt[1], Franziska U. Schwarzkopf[1], Siren Rühs[1], and Arne Biastoch[1, 2]

[1]GEOMAR Helmholtz Centre for Ocean Research Kiel, Kiel, Germany
[2]Christian-Albrechts-Universität zu Kiel, Kiel, Germany

**Correspondence:** Arne Biastoch (abiastoch@geomar.de)

**Abstract.** The inflow of relatively warm and salty water from the Indian Ocean into the South Atlantic via Agulhas leakage is important for the global overturning circulation and the global climate. In this study, we analyse the robustness of Agulhas leakage estimates as well as the thermohaline property modifications of Agulhas leakage south of Africa. Lagrangian experiments with both the newly developed tool Parcels and the well established tool Ariane were performed to simulate Agulhas leakage in

the eddy-rich ocean-sea ice model INALT20 (1/20° horizontal resolution) forced by the JRA55-do atmospheric boundary conditions. The average transport, its variability, trend and the transit time from the Agulhas Current to the Cape Basin of Agulhas leakage is simulated comparably with both Lagrangian tools, emphasising the robustness of our method. Different designs of the Lagrangian experiment alter in particular the total transport of Agulhas leakage by up to 2 Sv, but the variability and trend of the transport is similar across these estimates. During the transit from the Agulhas Current at 32°S to the Cape Basin, a

cooling and freshening of Agulhas leakage waters occurs especially at the location of the Agulhas Retroflection, resulting in a density increase as the thermal effect dominates. Beyond the strong air-sea exchange around South Africa, Agulhas leakage warms and salinifies the water masses below the thermocline in the South Atlantic.

## 1 Introduction

The Agulhas Current system is a key component of the global ocean circulation because it facilitates the exchange of water

between the Indian Ocean and the South Atlantic. The Agulhas Current is a strong western boundary current in the Indian Ocean, which flows southwards along the coast of Africa before turning back into the Indian Ocean as Agulhas Return Current (Lutjeharms, 2006). During this retroflection process, relatively warm and saline water from the Indian Ocean leaks into the South Atlantic by the shedding of large, anticyclonic Agulhas Rings, cyclonic eddies and direct inflow, commonly referred to as Agulhas leakage. Numerical studies indicate that the non-eddy flow contributes around half of the Agulhas leakage trans-

port, whereas Agulhas Rings are less important than previously assumed (Loveday et al., 2015; Cheng et al., 2016). Agulhas leakage supplies the upper limb of the Atlantic Meridional Overturning Circulation (AMOC) with a direct influence on the climate (Weijer et al., 2002; Donners and Drijfhout, 2004; Beal et al., 2011). Modelling studies suggest an increase in Agulhas leakage transport in recent decades (Biastoch et al., 2009; Rouault et al., 2009), but there is still no clear evidence of a long-term trend in Agulhas leakage from observations (Backeberg et al., 2012; Le Bars et al., 2014). An increase in Agulhas



leakage can be related to changes in Southern Hemisphere westerlies, especially their strengthening, under global warming (Durgadoo et al., 2013; Biastoch and Böning, 2013; Biastoch et al., 2015; Cheng et al., 2018). A sensitivity study using an idealised Lagrangian analysis set-up with a realistic eddy-rich ocean-sea ice model shows that such an increase in Agulhas leakage also increases the contribution of waters entering the South Atlantic from the Indian Ocean to the upper limb of the AMOC, while the contribution entering from the Pacific through Drake Passage decreases (Rühs et al., 2019).


The region south of Africa where the interbasin exchange occurs also is a region of vigorous thermohaline property modifications of Agulhas leakage waters by air-sea fluxes and mixing with other water masses (Boebel et al., 2003). Both mesoscale and submesoscale dynamics affect the thermohaline structure of upper and intermediate waters locally with different dynamical regimes in the vertical (Capuano et al., 2018). During the advection of Agulhas leakage waters into the South Atlantic,

the relatively warm upper water masses undergo cooling due to strong heat loss (Walker and Mey, 1988). The intermediate water of Agulhas leakage, Antarctic Intermediate Water (AAIW) from the Indian Ocean (I-AAIW), experiences mixing with the fresher AAIW of the South Atlantic (A-AAIW) in the Cape Basin to form the new Indo-Atlantic AAIW variety (Rusciano et al., 2012). Most of the studies on the evolution of thermohaline properties have, however, focused on a particular Agulhas Ring (e.g. Schmid et al., 2003) or specific region like the Cape Basin (Rimaud et al., 2012; Rusciano et al., 2012). One aim

of this study is to evaluate the modifications of the thermohaline properties of Agulhas leakage during the transit from the Agulhas Current into the Cape Basin and to identify the dominant effect on density associated with these modifications.

Several methods exist to estimate the Agulhas leakage transport in both the real ocean and using numerical models. First estimates were only built on the observations of Agulhas Rings using satellite data and in situ measurements based on the

assumption that Agulhas Rings carry the majority of Agulhas leakage (de Ruijter et al., 1999). Using subsurface floats and surface drifters, a mean Agulhas leakage transport including all varieties in which leakage can occur was estimated to be 15 Sv (1 Sv:= $10^6$ m$^3$s$^{-1}$) in the upper 1000 m (Richardson, 2007) and $21.3 \pm 4.7$ Sv in the upper 2000 m (Daher et al., 2020). Furthermore, a time series of the anomalies of Agulhas leakage transport was derived using satellite altimetry (Le Bars et al., 2014). It is not possible to determine the Agulhas leakage transport precisely from Eulerian velocity fields, even in

ocean models where three-dimensional velocity fields over time exist, since those also include the imprint of other circulation systems in the greater Agulhas region as for example the subtropical gyre in the South Atlantic (van Sebille et al., 2010). As a result, a Lagrangian approach or tracer based estimates in ocean models are widely used to analyse the variability, trends and characteristics of Agulhas leakage in more detail. For the tracer based estimate, either an additional passive tracer labelling Indian Ocean water masses is introduced (Loveday et al., 2015) or a passive tracer is calculated based on thermohaline

properties (Putrasahan et al., 2015). The tracking of Lagrangian particles is the most widely used approach to estimate Agulhas leakage in ocean models (e.g. Doglioli et al., 2006; Biastoch et al., 2008; van Sebille et al., 2009). This kind of Lagrangian leakage estimation always consists of releasing virtual fluid particles in the Agulhas Current representing a certain portion of the original transport, following their pathways within the simulated flow, and finally sampling those particles that enter and remain in the South Atlantic and hence sum up to Agulhas leakage. However, the employed Lagrangian tools and the



specific design of the Lagrangian experiments vary among studies. To assess the robustness of Lagrangian leakage estimates with respect to these differences is another aim of this study.

Most commonly, the Lagrangian tools Ariane (Blanke and Raynaud, 1997; Blanke et al., 1999) and the Connectivity Modeling System (CMS; Paris et al., 2013) are used to estimate Agulhas leakage. Ariane, because of its close correspondence to the numerical grids of NEMO (Madec and the NEMO Team, 2014) and ROMS (Shchepetkin and McWilliams, 2003), was used for example by Speich et al. (2006), Doglioli et al. (2006), Biastoch et al. (2008) and Durgadoo et al. (2013) and CMS was used for example by Weijer and van Sebille (2014), Biastoch et al. (2015), Cheng et al. (2018) and Daher et al. (2020). The transport of Agulhas leakage was only calculated once with the newly developed tool Parcels (Lange and Sebille, 2017; Delandmeter and van Sebille, 2019) and then compared to an estimate using CMS where a too strong diffusion of the velocity fields due to the interpolation method in CMS was suspected and therefore Parcels was preferred (Daher et al., 2020). In this study, we validate the Agulhas leakage estimates by Parcels against the results from Ariane, as Parcels is a more flexible Lagrangian tool, for example with respect to the layout of velocity field discretisation, and can be used for a variety of studies on Agulhas leakage dynamics in the future.

In addition, the design of the Lagrangian experiments varies in terms of the release and sampling sections, number of crossings of the control section and the date that the Agulhas leakage transport is assigned to. Lagrangian particles are released in the Agulhas Current either at 32°S or along the section of the Agulhas leakage Time-Series (ACT; Beal et al., 2015) at 34°S, which alters the mean Agulhas leakage transport, but does not affect its variability strongly (Cheng et al., 2016). In all studies, particles crossing an approximation of the Good Hope section (Ansorge et al., 2005) are referred to as Agulhas leakage. Although the Good Hope section is defined differently, it always lies west of the Agulhas Retroflection and cuts through the Cape Basin. Thereby, a crossing should ensure that waters with an Indian Ocean origin remain in the South Atlantic. Due to the turbulent dynamics in the Cape Basin, the Good Hope section might be crossed several times if a particle is trapped inside an eddy. An odd number of crossings can therefore be required to guarantee the remaining in the Atlantic (Cheng et al., 2016; Daher et al., 2020). A detailed comparison of the different parameters of the Lagrangian experiment is needed within the same ocean model to examine their effect on the resulting estimate of Agulhas leakage independent of the underlying circulation.

In this study, we estimate the transport and analyse the water mass characteristics of Agulhas leakage in an eddy-rich ocean-sea ice model (1/20° horizontal resolution) with two Lagrangian tools. First, we validate the simulated Agulhas leakage by the newly developed tool Parcels against the results by the well established tool Ariane and investigate the impact of different designs of the Lagrangian experiment in Sect. 3.1-3.3. In Sect. 3.4, the water mass characteristics of Agulhas leakage and their modifications south of Africa are analysed.



## 2 Materials and methods

### 2.1 Ocean model simulation

Output from a hindcast simulation with the eddy-rich ocean-sea ice model configuration INALT20 (Schwarzkopf et al., 2019) was used to conduct offline Lagrangian experiments. INALT20 is based on the ocean general circulation model NEMO (version 3.6; Madec and the NEMO Team, 2014) coupled to the Louvain-La-Neuve sea ice model version 2 (LIM-V2; Fichefet and Maqueda, 1997). A tripolar Arakawa C-grid with a horizontal resolution of 1/4° is used for the global host in which a nest with a horizontal resolution of 1/20° between 63°S–10°N and 70°W–70°E is embedded via two-way nesting using AGRIF (Adaptive Grid Refinement in Fortran; Debreu et al., 2008). In the vertical, INALT20 consists of 46 z-levels with a layer thickness of 6 m near the surface increasing to 250 m in the deepest layers. Cells at the bottom are allowed to be partially filled for a better representation of the bathymetry.

The hindcast simulation used here mainly differs from the experiment described and evaluated by Schwarzkopf et al. (2019) in the applied surface forcing. Here, atmospheric boundary conditions are given by the JRA55-do forcing data set covering the period from 1958 to 2019 (Tsujino et al., 2018). The main difference in this forcing set, compared to the previously applied COREv2 data set (Coordinated Ocean-Ice Reference Experiments data set version 2; Large and Yeager, 2009) is an increased horizontal (from 2° to 0.5°) and temporal (from 6 hourly, daily and monthly, depending on the forcing variable, to 3 hourly) resolution as well as the extension throughout the 2010s. Additionally, river runoff, which has previously been prescribed as a climatological field, is now interannually varying at daily resolution. Furthermore, 11 of the major tidal components are simulated here, while the experiment presented by Schwarzkopf et al. (2019) was non-tidal. All other parameters, including the preceding spin-up, are identical to their experiment "INALT20 NS_RW". Although there are differences between these simulations, the large scale horizontal circulation and also the representation of mesoscale variability in the area of interest to the present study are qualitatively comparable.

### 2.2 Lagrangian experiments

For the comparison of the two Lagrangian methods, Ariane and Parcels, different sets of Lagrangian experiments were conducted and the mean transport, interannual variability and trend from 1958 to 2014 of Agulhas leakage were analysed. In these experiments, particles were released in the Agulhas Current and all particles crossing the approximated Good Hope section in the Cape Basin were referred to as Agulhas leakage (Fig. 1a). To advect the particles, the three-dimensional 5-day mean velocity fields from INALT20 were used. This temporal resolution of the velocity fields is sufficient for these offline Lagrangian experiments as flow characteristics of the Agulhas Current system do not change significantly when using up to 9-day mean velocity fields (Qin et al., 2014).

In Ariane (Blanke and Raynaud, 1997; Blanke et al., 1999), virtual particles are advected along analytically computed streamlines, which are calculated from the three-dimensional, non-divergent velocity field on an Arakawa C-grid for each output time period. The velocities are linearly interpolated in space, while they are assumed to be constant in time for the given




output frequency of the model simulation. The interpolation and integration methods employed by Ariane respect the local non-divergence of the input files and the resulting trajectories represent volume transport pathways, which may experience

along-track tracer and density changes that reflect the sub-grid-scale parametrisations of the underlying ocean model. In our Lagrangian experiment with Ariane (version 2.3.0_02), which is hereafter referred to as experiment A, particles with an initial maximum transport of 0.1 Sv were released automatically and continuously according to the transport in the Agulhas Current at 32°S over 1 year and advected forward in time for 4 years. The transport assigned to a particle takes the reduced volume of partially filled cells at the bottom into account. In grid cells with a transport smaller than the maximum transport per particle,

1 particle per grid cell was seeded on the v-point at the centre of the 5-day mean model output fields (Fig. 1b). If the transport through a grid cell was greater than the maximum transport per particle, 8 particles were released, with 4 of them at the first quarter of the temporally-averaged model output and 4 particles after the third quarter. In our case, the former related to a release at 6 am the day before the centre of the 5-day mean output and the latter to a release at 6 pm the day after the centre of the 5-day mean output. Such a group of 4 particles was seeded regularly distributed in the x-z-plane centred around the

v-point. In partially filled cells at the bottom, particles were also released in the middle of the cell in the vertical and hence above particles at the same depth level in fully filled cells. A total of 57 of these 1-year release experiments with on average 186,000 particles per year were conducted for the years 1958-2014. As the experiments were run in the so called "quantitative" mode, the position, time and transport of a particle was only stored at its release as well as when the particle crossed a predefined section for the first time. All particles not leaving the domain enclosed by these sections within the integration time

were considered as "lost".

In Parcels (Lange and Sebille, 2017; Delandmeter and van Sebille, 2019), virtual particles are advected numerically with different schemes and customisable advection methods being available. If the velocity and tracer fields are discretised on a C-grid, the velocity fields are linearly interpolated in space and time, while the tracer fields are linearly interpolated in time, but

constant over a grid cell. In our first Lagrangian experiment with Parcels (hereafter referred to as experiment P), particles were 3-dimensionally advected forward in time using the 4th-order Runge-Kutta scheme with Parcels version 2.2.0. Although this integration scheme is not necessarily volume conserving, it has been used to estimate volume transport pathways (van Sebille et al., 2012, 2014) since without diffusion trajectories calculated with the 4th-order Runge-Kutta scheme and an appropriate time step are very similar to trajectories of the analytical, volume conserving method (van Sebille et al., 2018). When using

the 4th-order Runge-Kutta scheme, the time step $dt$ should be small enough that particles do not skip grid cells and thereby miss characteristic oceanographic features. After considering the trade-off between the accuracy of the time stepping and the computation time of the experiment, the timestep $dt$ was chosen such that particles stay in one model grid cell for at least 2 time steps. Therefore, a time step $dt$ of 16 minutes was calculated following $dt < ds_{min}/(v_{max} \cdot 2)$ where $ds_{min}$ is the smallest edge of all grid cells in the domain and $v_{max}$ is the largest horizontal velocity. The time step should also not be smaller than

necessary to avoid the accumulation of round-off errors. Similarly to experiment A, particles were released in the Agulhas Current at 32°S over 1 year for the years 1958-2014 and then advected for 4 years. Thereby, potential temperature $\theta$ and practical salinity $S$ were sampled and together with the position of the particles stored daily. The number of particles seeded

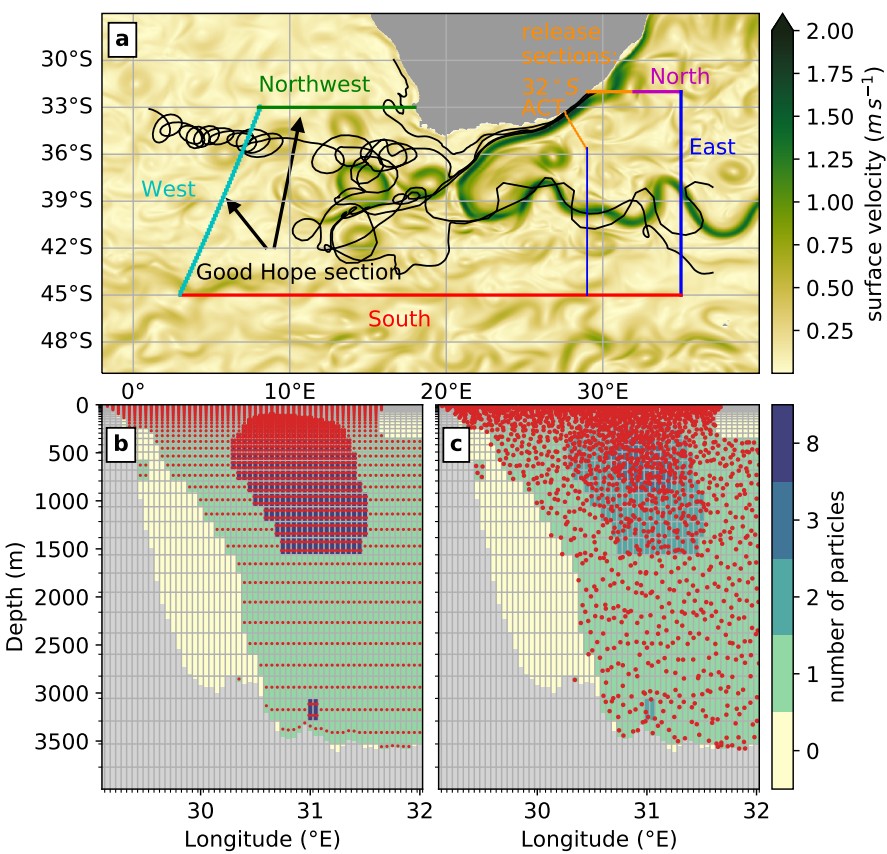

**Figure 1.** Design of the Lagrangian experiments A, P and P-ACT: (a) Release section 32°S for A and P and release section ACT for P-ACT in orange and all sampling sections as thick lines for A and P and as thin lines for P-ACT if differing from A and P. Exemplary trajectories of experiment P are shown as black lines as well as a snapshot of the surface velocity on 3 Jan 2010 as simulated in INALT20. Note that the sections "West" and "Northwest" are only an approximation of the locations of the observations at the Good Hope section. Release positions of particles in experiment (b) A and (c) P with a shading of the number of particles per grid cell. These particles were released at 12 pm on 3 Jan 2010 in experiment P and in experiment A if only 1 particle was released per grid cell. If 8 particles were released per grid cell in experiment A, 4 of them were released at 6 am on 2 Jan 2010 and the other 4 at 6 pm on 4 Jan 2010 from the same positions. In (b) and (c) the grid lines show the model grid and the grey shading the bathymetry with its partially filled grid cells at the bottom.

and their position was determined based on the transport of the Agulhas Current at 32°S. Therefore, the southward transport per grid cell at 32°S over the whole water column was calculated for each time step of the model output, i.e. every 5 days, and

the number of particles per grid cell and their individual transport identified accordingly such that the maximum transport per particle was 0.1 Sv. The calculated number of particles was seeded randomly in the x- and z-directions in each grid cell every 5 days (Fig. 1c) resulting in 110,000 particles being released on average each year. Although partially filled bottom cells are not implemented in Parcels, they were considered when computing the transport of the particles. While the release of the particles





in both experiments A and P are based on the same transport of the Agulhas Current at 32°S, the number of particles and their
individual transport as well as their release position and release time differ between A and P. The computational cost of those
experiments is also different with the runtime of experiment P being about 7 times longer than for A, but there is potential
to increase the model efficiency in Parcels with the aim of a fully parallel version (Delandmeter and van Sebille, 2019). The
section crossed by a particle was calculated in a second step such that only the first section crossed by a particle leaving the
domain was considered for later analysis.

For the comparison of the effect of different designs of the experiment, in particularly the release section, on the Agulhas
leakage transport, a second experiment was performed with Parcels (hereafter referred to as P-ACT). In P-ACT, the particles
were released along the ACT section at 34°S in the Agulhas Current (Fig. 1a). The section "East", used to define the transport
of the Agulhas Return Current, was thereby shifted westwards to 29°E. Otherwise, experiment P-ACT was conducted exactly
the same as experiment P.


In all experiments, the transport across the defined sections was calculated by summing up the transport of all particles cross-
ing the respective section and dividing by the number of release times per year. Only the section crossed first was considered
here as this is automatically done in experiment A. The Agulhas leakage transport is given by the sum of the transport of all
particles crossing the sections "West" and "Northwest", which approximate the Good Hope section.

## 2.3 Modifications of thermohaline properties

The modifications of the thermohaline properties of Agulhas leakage can be evaluated in terms of a horizontal distribution of
the density changes $d\rho$.

$$\frac{d\rho}{dt} = -\alpha(\overline{S}, \overline{\theta}, \overline{p})\frac{d\theta}{dt} + \beta(\overline{S}, \overline{\theta}, \overline{p})\frac{dS}{dt} + \kappa(\overline{S}, \overline{\theta}, \overline{p})\frac{dp}{dt} \tag{1}$$

where the overbar denotes a time average as follows

$$\overline{X} = \frac{X_{t+1} - X_t}{2} \tag{2}$$

$\alpha$ is the thermal expansion coefficient, $\beta$ the saline contraction coefficient, $\kappa$ the isentropic compressibility and $p$ pressure. As
we were only interested in the density changes due to the thermal and the haline effect, only the first and second term in Eq.
(1) were computed. First, the changes in density per day due to the thermal effect (Eq. (3)) and the haline effect (Eq. (4)) were
calculated for each particle $P$ along its trajectory. In a second step, all particles passing a certain geographical bin along their
way were selected and their local density changes per day weighted by the individual particle's transport $T$ were summed up.
Finally, this was divided by the cumulative sum over all particles of their individual transport multiplied by the length of each
particle's trajectory in days. Note that particles remaining in a bin longer than a day or crossing a bin several times were also





captured repeated times in the summation of the density changes as well as transports.

$$\frac{d\rho_\theta}{dt} = \frac{\sum_{j=lat_{min}}^{lat_{max}}\sum_{i=lon_{min}}^{lon_{max}}\sum_P -\alpha_P \frac{d\theta_P}{dt}T_P \mathbb{1}\{x_P \in [i, i+\Delta x)\}\mathbb{1}\{y_P \in [j, j+\Delta y)\}}{\sum_P T_P \mathbb{1}\{x_P \in \mathbb{Q}\}} \tag{3}$$

$$\frac{d\rho_S}{dt} = \frac{\sum_{j=lat_{min}}^{lat_{max}}\sum_{i=lon_{min}}^{lon_{max}}\sum_P \beta_P \frac{dS_P}{dt}T_P \mathbb{1}\{x_P \in [i, i+\Delta x)\}\mathbb{1}\{y_P \in [j, j+\Delta y)\}}{\sum_P T_P \mathbb{1}\{x_P \in \mathbb{Q}\}} \tag{4}$$

where $\mathbb{1}_A(x)$ is the indicator function

$$\mathbb{1}_A(x) := \begin{cases} 1 & \text{if } x \in A \\ 0 & \text{if } x \notin A \end{cases} \tag{5}$$

The bin size is $\Delta y = 2°$ in latitude $\times$ $\Delta x = 3°$ in longitude, which is larger than the maximum distance travelled per day by a particle of 206 km to avoid aliasing.

## 3 Results

### 3.1 Agulhas leakage transport

The mean (1958-2014) Lagrangian transport across all defined sections and its variability with respect to the release year agree well for experiments A and P (Fig. 2), despite the two different Lagrangian tools Ariane and Parcels being used and the resulting differences in advection schemes, interpolation and release strategy. In both A and P, the majority of the Agulhas Current transport at 32°S reflects back into the Indian Ocean and constitutes the Agulhas Return Current, thereby crossing the section "East" with a mean transport of 40.1 Sv in experiment A and 40.0 Sv in P. Particles with a combined transport of approximately 1 Sv accounting, for 2 % of the overall transport, do not leave the area within 4 years in both A and P. The Agulhas leakage transport for the years 1958 to 2014 is on average 9.7 Sv in experiment A and 9.9 Sv in P with a similar standard deviation in both experiments of 2.1 Sv. In general, we found a good agreement of the simulated transport across all sections in A and P with its differences between A and P being smaller than the standard deviation due to interannual variability. The simulated Agulhas leakage transport is, however, considerably lower than the estimate of mean Agulhas leakage from floats and drifters of at least 15 Sv in the upper 1000 m (Richardson, 2007) and 21.3 $\pm$ 4.7 Sv in the upper 2000 m (Daher et al., 2020). These observed transports of Agulhas leakage were obtained by weighting the ratio of leaking floats and drifters by the transport of the Agulhas Current at 32°S and ACT, respectively. Calculating the mean transport from 1994 to 2014 and hence for a similar period as the floats and drifters used by Daher et al. (2020), increases the mean Agulhas leakage transport in P only to 10.4 Sv.

A time series of the Agulhas leakage transport for A, P and P-ACT is shown in Fig. 2b. Here, the transport of the particles that were released in the Agulhas Current each year and then crossed the Good Hope section within 4 years were summed up





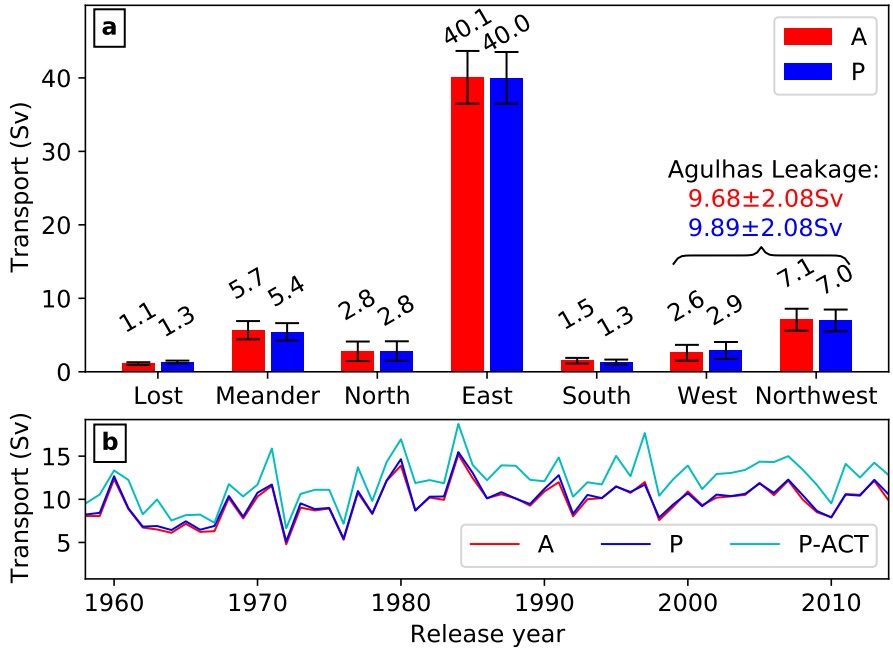

**Figure 2.** (a) Mean (1958-2014) transport across all sections as shown in Fig. 1 for experiment A in red and P in blue. The transport of all particles not crossing any section is shown as "Lost" and the transport of all particles leaving the region by crossing the release section again is shown as "Meander". (b) Time series of the Agulhas leakage transport for A (red), P (blue) and P-ACT (cyan).

for each release year. In all Lagrangian experiments, an upward trend in Agulhas leakage is only present between the beginning
of the 1960s towards the mid 1980s, but there is no clear trend afterwards. Depending on the exact period within the early 1960s
to mid 1980s the linear trend in Agulhas leakage is between 1 Sv and 2.5 Sv per decade with the linear trend being calculated
for all combinations starting in 1960-1962 and ending in 1982-1983. When considering all years from 1958 to 2014, the linear
trend per decade is only 0.5 Sv in both A and P (Fig. 2b). The lack of a clear upward trend in Agulhas leakage after the 1980s
is in contrast to earlier studies using hindcast simulations under the COREv2 forcing. Compared to Agulhas leakage estimates
using the same model configuration, but applying the COREv2 forcing (Schwarzkopf et al., 2019), where the linear trend in
Agulhas leakage transport from 1965 to 2000 is 1.1 Sv per decade, the experiments A and P exhibit a weaker trend of only
0.8 Sv per decade in the same period. This results in a lower mean Agulhas leakage transport for the period 1995-2005 of
10.4 Sv in both A and P compared to the estimate of 12.9 Sv for INALT20 under COREv2 forcing by Schwarzkopf et al.
(2019). In COREv2, the strengthening of the Southern Hemisphere westerlies is stronger compared to JRA55-do (Patara et al.,
2021) and a link was suggested between the increase in Agulhas leakage and the changes in Southern Hemisphere westerlies,
especially their strengthening (Durgadoo et al., 2013; Biastoch and Böning, 2013; Biastoch et al., 2015; Cheng et al., 2018).
Furthermore, first sensitivity studies indicate that the presence of tides in the experiment we analyse here, in contrast to the
non-tidal simulation under COREv2 forcing, might contribute to a weaker mean Agulhas Leakage. Understanding the details





of the impact of Southern Hemisphere westerlies and tides on Agulhas leakage and the differences between Agulhas leakage

in INALT20 forced with either JRA55-do or COREv2 is beyond the scope of this and subject to a dedicated study.

If a daily timeseries of Agulhas leakage is created by summing up the transport of particles that cross the Good Hope section each day, the transport varies considerably on seasonal timescales, which can be attributed to the passage of Agulhas Rings (not shown). The high correlation between A and P of these daily timeseries (r = 0.85, p < 0.01) furthermore underlines the robustness of our method in terms of specific leakage events with a transport of up to 10 Sv.

The latitude at which particles in the Lagrangian experiments are released in the Agulhas Current changes the mean Agulhas leakage transport, but it has a minor effect on the variability of Agulhas leakage (Cheng et al., 2016). The mean (1958-2014) Agulhas leakage transport in P-ACT, where particles were released along the ACT section at 34°S in the Agulhas Current, is 12.2 Sv and therefore 2.3 Sv (23 % of the Agulhas leakage transport in P) higher than in P with a release further north at 32°S (Fig. 2b). This can be partly explained by the increase in the transport of the Agulhas Current between these latitudes

due to the increase in Sverdrup transport (Beal et al., 2015). Both time series show a comparable interannual variability with a correlation of r = 0.92 (p < 0.01). A similar comparison of the release location in a climate model also revealed a good correlation between the Agulhas leakage transport time series, but an even stronger increase of the mean Agulhas leakage transport of 2.8 Sv (30 %) in the experiment with a release at ACT was found (Cheng et al., 2016). This agrees surprisingly well with our findings given that the atmospheric forcing of the ocean differs due to the intrinsic variability in a climate model

leading to a different temporal evolution and variability of Agulhas leakage. The strong nonlinearity of the southern Agulhas Current results in different degrees of recirculation south of ~34°S and hence the Agulhas Current transport in different models usually agrees at 32°S, but disagrees at ACT (Biastoch et al., 2018). The lower Agulhas leakage transport in P compared to the estimate from observations by Daher et al. (2020) can hence be partly explained by the different reference sections through the Agulhas Current.

## 3.2   Transit times from the Indian Ocean into the South Atlantic

The transit times of waters from the Agulhas Current at 32°S to the Good Hope section vary strongly from less than a month up to several years, where a maximum of 4 years is the upper limit due to the design of the experiments (Fig. 3a). The most frequent transit time (modal value of the transit time distribution) is 160 to 170 days for both A and P. The majority of transport (>50 %) reaches the Good Hope section within 280 days in A and 300 days in P. More particles with transit times of less

than 290 days are advected towards the Good Hope section in A compared to P, whereas there are more particles in P with transit times between 290 days and around 2 years than in A. The faster transit times in A might be due to the different temporal interpolation schemes in Ariane and Parcels (see Section 2.2). The transit time required to reach the Good Hope section for the majority of particles has a similar interannual variability and shows no significant trend in both A and P (not shown). The distribution of transit times from the Agulhas Current at 32°S towards 20°E is much narrower and shifted towards

shorter timescales (Fig. 3b) because of shorter and more direct pathways, where particles are mostly advected with the Agulhas Current. Therefore, 10 to 20 days is the most frequent transit time, the majority of transport reaches 20°E within 70 days and within 1 year 92 % of the transport arrives in the Atlantic at 20°E. In the Agulhas Current the flow is characterised by small



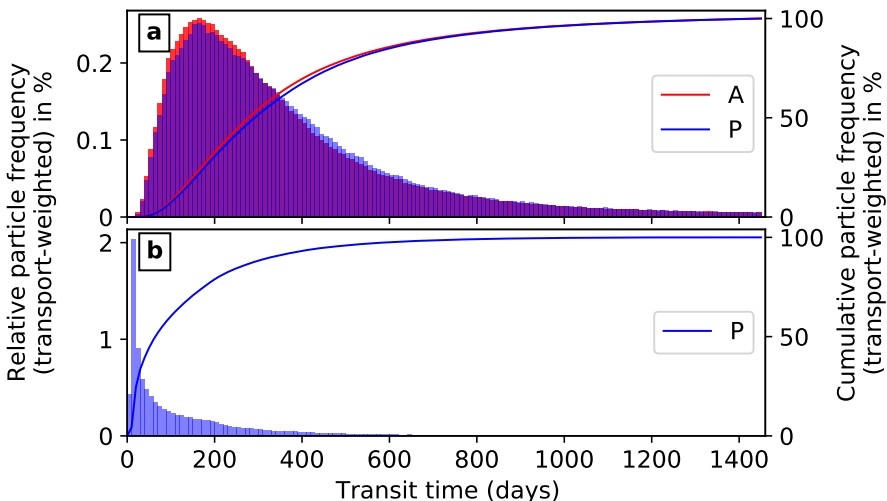

**Figure 3.** Mean (1958-2014) transit time distribution (transport-weighted) of Agulhas leakage from the Agulhas Current at 32°S to (a) the Good Hope section and (b) 20°E for experiment A in red and P in blue, overlapping probabilities are shown in purple. The transit time to 20°E can not be analysed in experiment A because it was run in the quantitative mode of Ariane.

differences in velocity and weak recirculation, but once the waters of Agulhas leakage are in the Atlantic, recirculation through interaction with and within the mesoscale in the Cape Basin occurs.

## 3.3   Robustness of Agulhas leakage estimates to different experimental designs

Different dates of reference to calculate a time series of the Agulhas leakage transport alter the transport per year, but not the decadal variability and trend of the Agulhas leakage transport. In previous studies, the date of reference of the Agulhas leakage time series has been either the year in which virtual particles were released in the Agulhas Current (e.g. Biastoch et al., 2009; Schwarzkopf et al., 2019) or the date of the last crossing of the Good Hope section (e.g. Loveday et al., 2015; Cheng et al.,
2016). An approximation of the Good Hope section has been used in all of these studies as the control section to ensure that waters with an Indian Ocean origin remain in the South Atlantic. The exchange of Indian Ocean and Atlantic water masses occurs already east of the Good Hope section south of the tip of Africa at 20°E, though. Therefore, the first crossing of 20°E is a good reference date for the Agulhas leakage time series as long as the remaining of the waters in the South Atlantic is ensured by the crossing of the Good Hope section afterwards. Due to the fast transit time from the release section in the Agulhas
Current at 32°S towards 20°E (Fig. 3b), the release year can be used as an approximation of the year of the 20°E crossing (Fig. 4a, r = 0.83, p < 0.01). For those two time series only the crossing of the Good Hope section was required to be counted as Agulhas leakage. Particles might, however, cross the Good Hope section several times and only particles with an odd number of crossings actually remain in the South Atlantic within the given time period. Applying this additional criterion to the time series referenced to the release year, leads to a reduced Agulhas leakage transport by 1 Sv on average, but with nearly the same





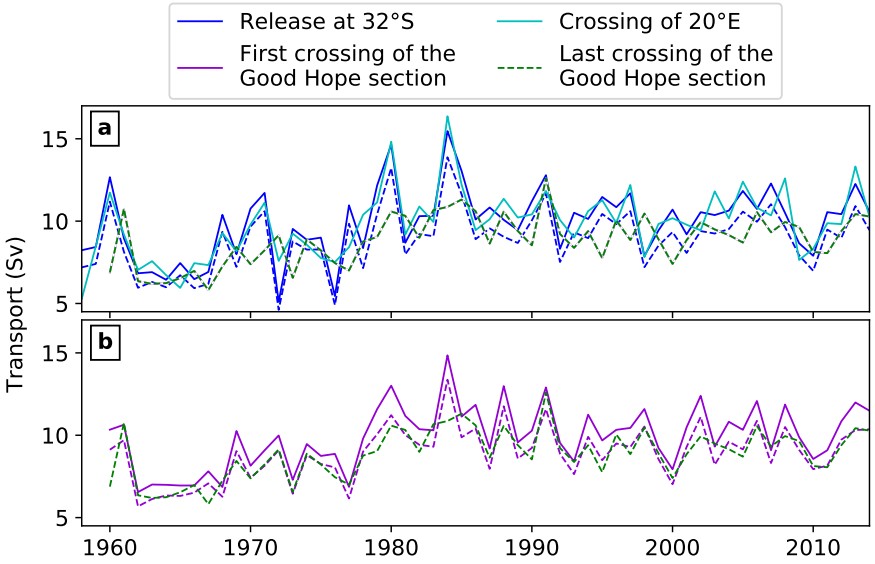

**Figure 4.** Comparison of the year of reference used to calculate a time series of the Agulhas leakage transport for experiment P: time series of Agulhas leakage transport referenced to the release year (blue in (a)), year of crossing 20°E (cyan in (a)), year of the first crossing of the Good Hope section (purple in (b)) and year of last crossing of the Good Hope section (green in (a) + (b)). Dashed lines show the combined transport of all particles that cross the Good Hope section an odd number of times, while for the solid lines this additional criterion was not applied.

interannual variability (r = 0.996, p < 0.01). The transport time series referenced to the release year and the year of the last crossing of the Good Hope section do not correlate well (r = 0.37, p < 0.05), which is expected due to the transit time. The highest correlation between those time series occurs when the transport time series referenced to the last crossing lags 1 year (r = 0.76, p < 0.01). Although the different samplings influence the interannual variability, the decadal variability is not affected and agrees across all time series. Using the date of the Good Hope section crossing as the reference for the Agulhas leakage

time series enables to capture crossings of e.g. Agulhas Rings. There is not a big difference between the Agulhas leakage time series referenced to the first or last crossing of the Good Hope section (r = 0.84, p < 0.01). Not applying the additional criterion of an odd number of crossings on the time series referenced to the first crossing results again in a higher average transport and only minor changes of the variability.

The date of reference does not have a strong effect on the trend of the Agulhas leakage transport. The trend of the time series

is 0.46 Sv per decade in 1958-2014 when referenced to the last crossing of the Good Hope section as well as when the release year is used as a reference. Due to the potential impact of an increasing leakage under global warming on the AMOC (Beal et al., 2011; Biastoch et al., 2015), the overall trend of the Agulhas leakage transport might be more important than the Agulhas leakage transport in a particular year. As a general recipe, we propose to use the year of release as reference for the Agulhas leakage transport time series calculated from the particles crossing the Good Hope section within 4 years. This represents the



mean exchange between the Indian and Atlantic Ocean and its temporal changes on decadal timescales towards long-term
trends well while keeping the Lagrangian experiment as simple as possible by not introducing another control section at 20°E.
Whether all particles crossing the Good Hope section or only those with an odd number of crossings should be considered
depends on the research question. If the advective effect of Agulhas leakage on the overturning circulation is of interest, only
particles crossing the Good Hope section an odd number of times and thereby remaining in the South Atlantic shall be referred

to as Agulhas leakage. On the other hand, all particles that cross the Good Hope section are important when analysing the effect
of Agulhas leakage on the thermohaline properties of water masses in the Southeast Atlantic as mixing with water masses from
the Atlantic occurs for example already in the Cape Basin (Rimaud et al., 2012; Rusciano et al., 2012), which is so turbulent
that particles can either stay there longer than the integration time of the Lagrangian experiment or escape and eventually join
the Agulhas Return Current.

### 310 3.4 Thermohaline property modifications of Agulhas leakage waters

In the following, the thermohaline characteristics of Agulhas leakage are analysed for experiment P only as we showed that
the average transport, its variability and the transit time of Agulhas leakage agree well between experiments A and P. Here, all
particles crossing the Good Hope section, regardless of the number of crossings, are analysed.

On its transit into the Atlantic Ocean, an overall cooling and freshening of Agulhas leakage occurs especially at the location

of the Agulhas Retroflection, but the modification of thermohaline properties differs between water masses. The $\theta S$-diagram
in Fig. 5 reveals that nearly half of the Agulhas leakage transport at both 32°S in the Agulhas Current and at the Good Hope
section consist of central waters with potential density anomalies referenced to the sea surface ($\sigma_0 = \rho_0 - 1000$ kg m$^{-3}$; in
the following the units are dropped for better readability) of 26 to 26.9. This agrees with in-situ observations showing that
the core of Agulhas Rings consists of a transformed variety of South Indian Central Waters and Subtropical Mode Waters

with $26 < \sigma_0 < 26.8$ (Arhan et al., 1999, 2011). During the transit from the Agulhas Current (Fig. 5a) towards the Good
Hope section (Fig. 5b) the amount of upper waters ($\sigma_0 < 26$) reduces by 0.9 Sv to 2.0 Sv, equivalent to a reduction of 30 %.
Thereby, the waters with $\sigma_0 < 25$ are mostly eroded by cooling due to very strong surface heat fluxes in this region (Walker
and Mey, 1988). At the same time, the amount of central waters increases by 0.5 Sv to 5.0 Sv and the amount of AAIW
($26.9 < \sigma_0 < 27.5$) increases by 0.3 Sv to 2.6 Sv. As more than 95 % of the Agulhas leakage transport is within the upper

1200 m, deep waters denser than $\sigma_0 = 27.5$ constitute a negligible amount of Agulhas leakage in agreement with observations
(Gordon et al., 1987). This confirms the assumption by Richardson (2007) and Daher et al. (2020) that available floats drifting
at 1000 m represent Agulhas leakage over the full depth well.

The modifications of temperature and salinity and their effect on density are shown in more detail in Fig. 6 and 7 for each
water mass. The upper waters and the AAIW cool continuously during the transit, while the temperature distribution of the

central waters broadens especially between 32°S and 20°E (Fig. 6a, d, g). The horizontal distribution of density changes due to
the thermal effect in the left column of Fig. 7 confirms the cooling of all water masses, leading to a density gain, which occurs
in most of the area south of Africa. The upper waters experience the strongest density gain due to cooling, which occurs in the
Agulhas Current and the southern part of the Cape Basin within the Agulhas Ring corridor. A density loss takes place in the



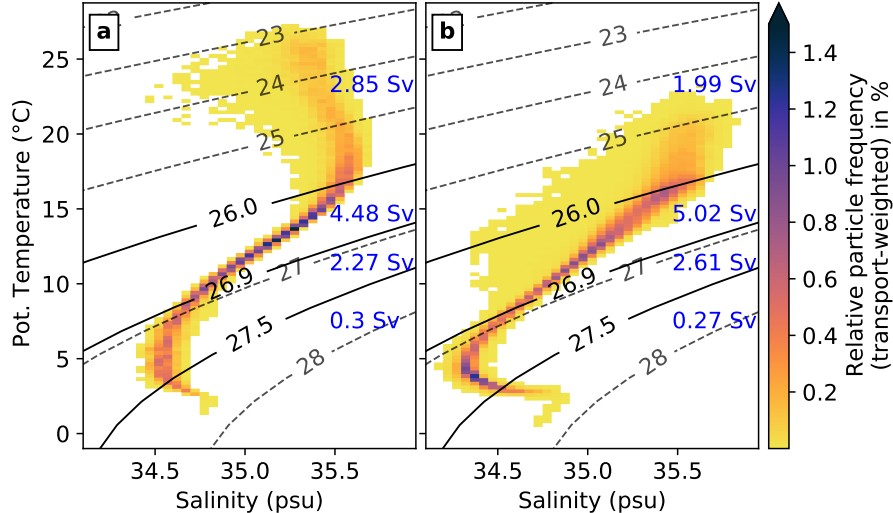

**Figure 5.** $\theta S$-diagram of Agulhas leakage for experiment P (a) in the Agulhas Current at 32°S and (b) at the first crossing of the Good Hope section. The relative transport-weighted particle frequency is shown per 0.25°C×0.05 psu bin in percent. Potential density levels used to separate upper, central, intermediate and deep waters are highlighted by solid black lines and blue numbers indicate the transport of these water masses.

northern part of the Cape Basin, but of smaller magnitude compared to the gain. The area of the strongest density changes due
to the thermal effect is within the major pathway of Agulhas leakage around South Africa following the Agulhas Current and
the Agulhas Ring corridor (Fig. 7a). The density changes of the central waters and the AAIW have a similar pattern, but they
have a smaller magnitude for the AAIW, with a density decrease due to warming along the coast of South Africa and a density
increase in particular around the Agulhas Retroflection. The mean depth of the $\sigma_0$=26 surface, which is the upper boundary
of the central waters, suggests that different dynamical processes are responsible for the density changes of the central waters
south of Africa. In the Agulhas Current and Agulhas Retroflection region central waters are mostly found below at least 100
m, which is below the mean winter mixed layer depth (indicated by the cyan line in Fig. 7f), suggesting a change in density
mainly due to mixing with other water masses. To the northwest, in particular in the Cape Basin, the depth of the central waters
shallows to a minimum depth of 50-100 m, which is within the mean winter mixed layer, and therefore air-sea fluxes might
modify the thermohaline properties of the shallower central waters. The modifications of the salinity has a smaller effect on the
density compared to the thermal effect, but is not negligible especially for the intermediate waters. There is a salinification and
hence density gain of the upper waters all around South Africa with freshening only in the southern part of the Cape Basin (Fig.
7e). This is in agreement with a slight shift towards higher salinities between the release and 20°E and a decrease afterwards
in Fig. 6b. The central waters and the AAIW experience a freshening, which dominates in the region and partly cancels out the
thermal effect on the density. The freshening of AAIW is a result of mixing with AAIW from the Atlantic in the Cape Basin
(Rimaud et al., 2012; Rusciano et al., 2012). The changes in density of all water masses of Agulhas leakage combined shows

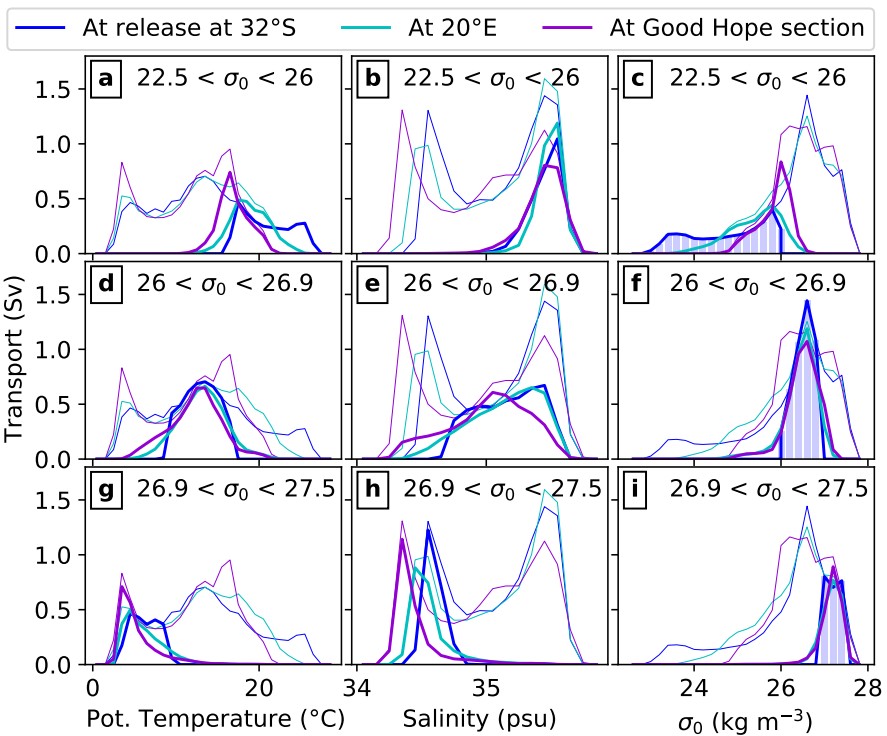

**Figure 6.** Thermohaline property modification of Agulhas leakage waters for experiment P: Transport of (a-c) upper waters ($22.5 < \sigma_0 < 26$), (d-f) central waters ($26 < \sigma_0 < 26.9$) and (g-i) AAIW ($26.9 < \sigma_0 < 27.5$) per potential temperature (a, d, g, 1°C bins), salinity (b, e, h, 0.1 psu bins) and $\sigma_0$ (c, f, i, 0.2 kg m$^{-3}$ bins) class. To assign waters to these water masses, $\sigma_0$ in the Agulhas Current at 32°S is used, highlighted by the blue shading. The properties at the release at 32°S are shown as blue, at 20°E as cyan and at the first crossing of the Good Hope section as purple line. The thin lines show the transport of all water masses together including the deep waters ($\sigma_0 > 27.5$).

that cooling in the Cape Basin has the strongest impact on the density (Fig. 7b). Initially Agulhas leakage waters are relatively warm and saline, which is a density compensated anomaly. As they lose their thermal signature during the transit, a positive density anomaly due to the anomalously high salinity spreads north into the South Atlantic (Weijer et al., 2002; Biastoch et al., 2009).

**4 Conclusions**

In this study, we assessed the robustness of Agulhas leakage estimates as well as the thermohaline property modifications of Agulhas leakage between the Agulhas Current and the Cape Basin in the eddy-rich ocean-sea ice model INALT20 (1/20° horizontal resolution). We performed Lagrangian experiments to estimate Agulhas leakage, validate the results by the Lagrangian tool Parcels against the one by the well established tool Ariane and analyse the effect that different designs of the Lagrangian

experiments have on the simulated transport. In these experiments, particles were released in the Agulhas Current and all par-

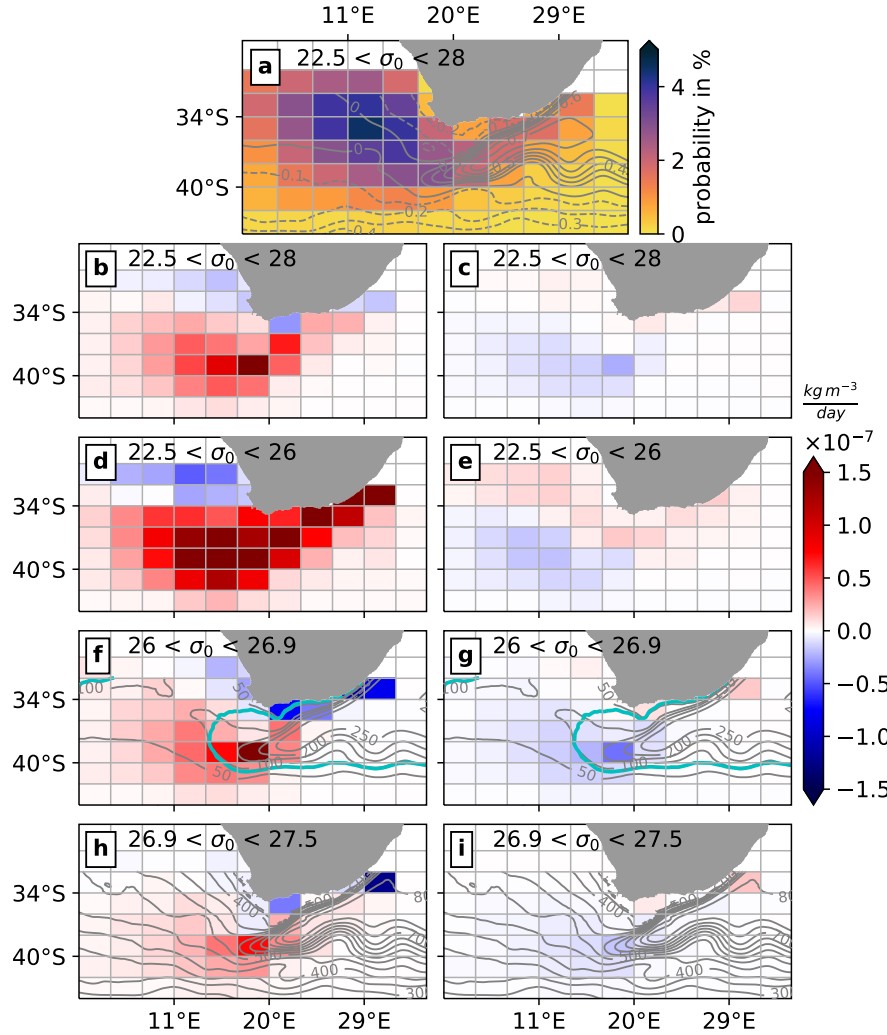

**Figure 7.** Pathways and horizontal distribution of the density changes of Agulhas leakage for experiment P: (a) Relative transport-weighted probability in percent of all Agulhas leakage waters (considering all Agulhas leakage particles per bin at each time step). Contour lines show the mean (1958-2014) sea surface height (in m). Density changes due to the (b, d, f, h) thermal and (c, e, g, i) haline effect for (b+c) all water masses, (d+e) upper waters ($22.5 < \sigma_0 < 26$), (f+g) central waters ($26 < \sigma_0 < 26.9$) and (h+i) AAIW ($26.9 < \sigma_0 < 27.5$). To assign waters to these water masses, $\sigma_0$ in the Agulhas Current at 32°S is used. The bin size of all histograms is 2° in latitude $\times$ 3° in longitude. See Sect. 2.3 for details on the calculation. Grey contour lines in (f+g) and (h+i) show the mean (1958-2014) depth of the $\sigma_0 = 26$ and $\sigma_0 = 26.9$ surfaces, respectively. East of the cyan line the mean (1958-2014) depth of $\sigma_0 = 26$ is below the mean (1958-2014) winter (JJA) mixed layer depth.

ticles crossing the Good Hope section in the Cape Basin were referred to as Agulhas leakage. The regions of the strongest modifications of the thermohaline properties during this transit were identified.



The mean (1958-2014) transport of Agulhas leakage and the Agulhas Return Current and their variability as well as the transit times and trend of Agulhas leakage agree in the experiments with Parcels and Ariane. We estimated a mean (1958-2014)

Agulhas leakage transport of 9.7 Sv with Ariane and 9.9 Sv with Parcels. The transit time of Agulhas leakage from the Agulhas Current at 32°S to the Good Hope section varies between less than a month to several years with the most frequent transit time of half a year. Agulhas leakage increases from the early 1960s to mid 1980s by 1 Sv to 2.5 Sv per decade depending on the exact period for the calculations, but we find no clear trend afterwards under the JRA55-do forcing, which is in contrast to previous studies using hindcast simulations under the COREv2 forcing. The causes of the different trends need further investigations,

but an explanation might be the more pronounced strengthening of the Southern Hemisphere westerlies in COREv2 compared to JRA55-do (Patara et al., 2021) as an increase in Agulhas leakage is linked to the changes in Southern Hemisphere westerlies, especially their strengthening (Durgadoo et al., 2013; Biastoch and Böning, 2013; Biastoch et al., 2015; Cheng et al., 2018).

The effect of different designs of the Lagrangian experiment on the Agulhas leakage transport is more pronounced than using the two different Lagrangian tools Parcels and Ariane. Shifting the release section in the Agulhas Current further south

to the ACT section at 34°S increases the mean transport of Agulhas leakage by 2.3 Sv to 12.2 Sv, but the interannual variability is comparable. Calculating the Agulhas leakage transport only from particles that cross the Good Hope section an odd number of times results in a decrease by 1 Sv. Different dates of reference to calculate a time series of the Agulhas leakage transport alter the transport per year, but not the decadal variability and trend of it. This, in combination with the fast transit time of the majority of Agulhas leakage waters of less than a year from the Agulhas Current at 32°S to the Good Hope section,

supports the use of the release year as the reference time for the Agulhas leakage transport. All of these test cases show that changing certain parameters of the Lagrangian experiment can affect in particular the total transport of Agulhas leakage and that designing such an experiment to estimate Agulhas leakage is not straight forward due to the turbulent regime. Comparing the total transport of studies with different ocean models, observational products and techniques to estimate Agulhas leakage needs to be done with caution, while transport anomalies and trends are assumed to be more robust across different estimates

if the same atmospheric conditions prevail. Furthermore, the horizontal resolution of the ocean model employed here is not sufficient to resolve submesoscale dynamics. Increasing the horizontal resolution to 1/60° in the INALT model family and thereby resolving submesoscale flows improves the representation of mesoscale eddies in the Agulhas Ring corridor (Schubert et al., 2019, 2020). In particular the representation of lee cyclones west of the Agulhas bank improves when submesoscale flows are resolved (Schubert et al., 2021, submitted manuscript). The lee cyclones are important for the formation of Agulhas

filaments, which contribute to 40 % more Agulhas leakage, when the resolution is increased to 1/60° (Schubert et al., 2021, submitted manuscript). The absence of submesoscale flows thus can partly explain the too low Agulhas leakage transport in INALT20 compared to observations. Submesoscale dynamics also influence the mode and intermediate water properties by the formation of mode waters in Agulhas Rings, which only occurs if the submesoscale is resolved, and the AAIW transformation in its regional varieties (Capuano et al., 2018).

In terms of thermohaline properties, we showed that they are modified not only in the Cape Basin, but also in other regions of the Agulhas Current system. Agulhas leakage consists of upper and intermediate waters with nearly half of Agulhas leakage being central waters at both 32°S in the Agulhas Current and at the Good Hope section. During the transit from the Agulhas



Current at 32°S to the Cape Basin the amount of upper waters decreases, whereas the amount of central waters and AAIW increases. The water property modification during the transit is due to a cooling and freshening of Agulhas leakage waters,

which occurs especially at the location of the Agulhas Retroflection. This leads to a density increase as the thermal effect dominates. The upper waters experience the strongest changes in density, especially due to a cooling through heat loss in the Agulhas Current and the Agulhas Ring corridor in the southern part of the Cape Basin. The freshening is more pronounced in the central waters and AAIW, while in the upper waters the salinification and freshening in different regions compensate each other. The impact of Agulhas leakage and its temporal changes can not only be interpreted based on the transport, but also the

thermohaline properties of Agulhas waters and their spatial and temporal changes need to be taken into account. A substantial increase in total heat and salt fluxes across the Good Hope section over the last decades is the result of a combined effect of changes in transport and water mass properties of Agulhas leakage (Loveday et al., 2015).

We have confirmed that estimating Agulhas leakage with the Lagrangian tool Parcels leads to very similar results as simulations with Ariane. Even though the numerical integration scheme of Parcels is not volume conserving by definition as the

analytical method in Ariane, our results show that Parcels can be used for volume transport estimations. This opens up new opportunities for future studies due to the flexibility and ongoing development of Parcels. Agulhas leakage can, for example, also be estimated with Parcels based on a variety of differently gridded products, e.g. reanalysis products, which do not have a non-divergent velocity field as required by Ariane. Velocity fields of an ocean model with a non-linear free surface can also not be used in Ariane. Another useful feature of Parcels is the ability to build a realistic, global velocity field consisting of model

output with different resolutions for different domains. In the case of INALT20, particles would be advected with velocities from the nest with its mesoscale resolving horizontal resolution of 1/20° in the Agulhas Current system and velocities from the global host (1/4° horizontal resolution) everywhere else. Such an experimental design can be used to for example analyse the far-field impact of Agulhas leakage on the AMOC. A disadvantage of Parcels at the moment is the longer computing time compared to Ariane, but the Parcels team is working on an increased efficiency and aims for a parallel version (Delandmeter

and van Sebille, 2019).

*Code and data availability.* The Lagrangian software Parcels is available at http://oceanparcels.org/ and Ariane at http://mespages.univ-brest.fr/~grima/Ariane/. For reproducibility of all results, the scripts to perform the Lagrangian experiments and the postprocessing scripts as well as all data required to produce the figures are made available through GEOMAR (https://hdl.handle.net/20.500.12085/b704e917-09dd-4a73-b6a1-ea24a549920c).

*Author contributions.* AB, SR and CS defined the overall research problem and methodology; FUS developed, ran and validated the ocean model simulation and performed the Lagrangian experiments with Ariane; CS performed the Lagrangian experiments with Parcels, analysed all Lagrangian simulations, produced all figures and prepared the paper. All coauthors discussed the analyses and contributed to the text.





*Competing interests.* The authors declare that they have no conflict of interest.

*Acknowledgements.* The ocean model simulation was performed at the North-German Supercomputing Alliance (HLRN) and on the ESM
partition at Jülich supercomputing centre (JUWELS); the trajectory simulations were conducted at the Christian-Albrechts-Universität zu
Kiel (NESH). We thank the Ariane and Parcels teams for support. This research has been supported by the German Federal Ministry of
Education and Research (grant no. SPACES-CASISAC (03F0796A)). It has also received funding from the Initiative and Networking Fund
of the Helmholtz Association through the project "Advanced Earth System Modelling Capacity (ESM)". We thank René Schubert for helpful
discussions and Willi Rath and Katharina Höflich for technical support.





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
