# Peer review of "Characteristics and robustness of Agulhas leakage estimates: an inter-comparison study of Lagrangian methods"

_Ocean Science, 2021_

## Referee Comment (RC1)

**Reviewer's comments**

Lagrangian particle experiments have been widely implemented to quantify Agulhas leakage. But there are many subtle nuances between different configurations (i.e., the tool of choice, frequency of particle release, the definition of leakage water, etc.) In a simulation of a 1/20 deg ocean model (INALT20), the authors compare the Agulhas leakage estimates and their variability using the widely-used Ariane tool to a modern and actively developed tool, Parcels, over a wide range of configurations. There are three major parts of the result: (1) Validate Parcels to Ariane (2) Experimenting various designs in Parcels, and (3) thermohaline characteristics sampled by trajectories using Parcels.

This work serves as a validation to the newly developed Parcels. It also addresses some common confusions of implementing Lagrangian experiments. Moreover, the discussion of Thermohaline changes of various water types along the Agulhas Leakage pathway is a great addition. The writing is of excellent quality with extensive references to the topic. Once some comments are addressed, I recommend accepting this paper.

**Comments:**

- It might be better to add a table to summarize all Lagrangian experiments/designs included. The readers have to go deep into the sentences to find the differences between Ariane, Parcels, Parcels-ACT, and the tests of different referencing dates.
- Perhaps add more lines to justify why only run Ariane in `quantitative mode` and why only doing Water Chacrateristic analyses in `Parcels`
- L.128. So particles are released over one year and advected for extra four years, maximum transit time four years, but full experiment length five years?
- L.129. Could you please elaborate on why the Release Strategy of Ariane has to be done in such a way? (four particles at the 1/4 of 5days, and another four at 3/4 of 5 days)? Why is this not necessary for Parcels?
- L. 160 I still don't fully understand how the number of release particles is determined in each box with a Maximum of 0.1 Sv. I assume something like this? if a grid box with transport <0.1 Sv, if say 0.35 Sv, release 3 * 0.1 + 5 * 0.01? Perhaps a concrete example here can help new users of Lagrangian tools.
- L. 188, not very clear to me how a "local density changes of each particle" is calculated. I assume it's the Density difference (from Potential Temperature and Salinity) divided by the time that particle crosses the bin. So this has to be calculated per bin/per particle?

- L. 191, why the sum of all particles has to be multiplied by the length of each particle's trajectories? What's in days? The "length of the trajectory," or the "cumulative sum of transport multiplied by the length of trajectory?"
- Fig. 2: why not include the P-ACT in the bar plot? It would be interesting to see if P-ACT at all other sections.
- I like the reasoning for choosing release year as the reference date, including the evidence of fast transition time to 20E, and the strong mixing between 20E and GH line.
- L. 405: I understand that thermohaline properties are equally important. Are there more references/citations of increasing heat/salt fluxes?
- L. 418: ",for example," It's nice to state the drawback/advantage of P/A and the future opportunities to use P, but it seems not to be a good idea to conclude this great work. Maybe some reorganization for the last paragraph.

---

## Author Response (AR1)

**Response to Anonymous Referee #1**

*Original reviewer's comments are inserted in black, Author Replies (AR) are added in blue, and Changes made to the Manuscript (CM) are finally listed in grey, whereby page and line numbers refer to the fully revised version of the manuscript.*

Lagrangian particle experiments have been widely implemented to quantify Agulhas leakage. But there are many subtle nuances between different configurations (i.e., the tool of choice, frequency of particle release, the definition of leakage water, etc.) In a simulation of a 1/20 deg ocean model (INALT20), the authors compare the Agulhas leakage estimates and their variability using the widely-used Ariane tool to a modern and actively developed tool, Parcels, over a wide range of configurations. There are three major parts of the result: (1) Validate Parcels to Ariane (2) Experimenting various designs in Parcels, and (3) thermohaline characteristics sampled by trajectories using Parcels. This work serves as a validation to the newly developed Parcels. It also addresses some common confusions of implementing Lagrangian experiments. Moreover, the discussion of Thermohaline changes of various water types along the Agulhas Leakage pathway is a great addition. The writing is of excellent quality with extensive references to the topic. Once some comments are addressed, I recommend accepting this paper.

AR: Thank you for your kind reply and constructive criticism below which helped to improve the manuscript.

Comments:

- It might be better to add a table to summarize all Lagrangian experiments/designs included. The readers have to go deep into the sentences to find the differences between Ariane, Parcels, Parcels-ACT, and the tests of different referencing dates.

  AR: Thanks for your suggestion. We included a table with parameters of the Lagrangian Experiments (Table 1). Regarding the different reference dates, we think this is best described visually. We therefore added another panel to the corresponding figure (Fig. R1 in this document) with a map indicating the sections used to assign the reference date of a particle.

  CM: ll.113-115 For the comparison of the two Lagrangian methods, Ariane and Parcels, different sets of Lagrangian experiments (Table 1) were conducted and the mean transport, interannual variability and trend from 1958 to 2014 of Agulhas leakage were analysed.

Table 1: Parameters of the Lagrangian Experiments.

| Lagrangian experiment | A | P | P-ACT |
|---|---|---|---|
| Lagrangian tool | Ariane | Parcels | |
| Advection method | analytic | 4th-order Runge-Kutta | |
| Release section | 32°S | | ACT at 34°S |
| Release positions | regularly | randomly | |
| Release frequency | 3× every 5 days over 1 year | once every 5 days over 1 year | |
| Release period | 1958-2014 | | |
| Advection time | 4 years | | |
| Particles released on average per year | ca. 186,00 | ca. 110,000 | ca. 176,000 |

[Figure]

Figure R1: Comparison of the year of reference used to calculate a time series of the Agulhas leakage transport for experiment P: time series of Agulhas leakage transport referenced to the release year (blue in (a)), year of crossing 20°E (cyan in (a)), year of the first crossing of the Good Hope section (purple in (b)) and year of last crossing of the Good Hope section (green in (a) + (b)). Dashed lines show the combined transport of all particles that cross the Good Hope section an odd number of times, while for the solid lines this additional criterion was not applied. (c) Schematic path of the Agulhas Current via the Retroflection into the Agulhas Return Current (black arrow) and all and all the sections used for assigning the reference date in (a) and (b). The Cape Basin is located northwestwards of the Agulhas Retroflection and southeastwards of the Good Hope section.

- Perhaps add more lines to justify why only run Ariane in quantitative mode and why only doing Water Characteristic analyses in Parcels.

  AR: Using the quantitative mode of Ariane is the established method to estimate Agulhas leakage, which we wanted to use as a reference for the validation of the experiment with Parcels. We also aimed to replicate the seeding strategy of the quantitative mode in Ariane in our Parcels experiments. The water characteristic analyses are not possible in the quantitative mode of Ariane, but as we showed that the experiments A and P compare well, doing the water characteristic analyses in P only seemed appropriate.

- L.128. So particles are released over one year and advected for extra four years, maximum transit time four years, but full experiment length five years?

  AR: Yes, the full length of an experiment is 5 years to be able to also advect the particles released last for 4 years. We included this detail.

  CM: ll.126-130 In our Lagrangian experiment with Ariane (version 2.3.0_02), which is hereafter referred to as experiment A, particles with an initial maximum transport of 0.1 Sv were released automatically and continuously according to the transport in the Agulhas Current at 32°S over 1 year. All trajectories were integrated forward in time over the same length of 4 years after initialisation, so that the total experiment length is 5 years.

- L.129. Could you please elaborate on why the Release Strategy of Ariane has to be done in such a way? (four particles at the 1/4 of 5days, and another four at 3/4 of 5 days)? Why is this not necessary for Parcels?

  AR: This seeding strategy is automatically used in the quantitative mode of Ariane and not a choice of ours. As we use 5-day mean fields for the advection of particles, a release every 5 days is sufficient and was therefore used in the experiments with Parcels.

  CM: ll.131-135 The built-in seeding strategy in the quantitative mode of Ariane is as follows: In grid cells with a transport smaller than the maximum transport per particle, 1 particle per grid cell is seeded on the v-point at the centre of the 5-day mean model output fields (Fig. 1b). If the transport through a grid cell is greater than the maximum transport per particle, 8 particles are released, with 4 of them at the first quarter of the temporally-averaged model output and 4 particles after the third quarter.

- L. 160 I still don't fully understand how the number of release particles is determined in each box with a Maximum of 0.1 Sv. I assume something like this? if a grid box with transport <0.1 Sv, if say 0.35 Sv, release 3 * 0.1 + 5 * 0.01? Perhaps a concrete example here can help new users of Lagrangian tools.

  AR: Thanks for suggesting to include such an example. The transport per grid box is, however, divided equally by the number of particles such that the transport assigned to all particles is the same.

  CM: ll.163-164 If the transport through a grid cell was e.g. 0.24 Sv, which can occur in the core of the Agulhas Current, 3 particles were released in that grid cell with each of them having a transport of $0.24 Sv/3 = 0.08 Sv$

- L. 188, not very clear to me how a "local density changes of each particle" is calculated. I assume it's the Density difference (from Potential Temperature and Salinity) divided by the time that particle crosses the bin. So this has to be calculated per bin/per particle?

  AR: What we meant with "local density changes of each particle" was the density change per day of each particle along its trajectory which was calculated in the first step. So this is calculated per particle and no binning has been done yet. The binning happens in a second step, which is explained in more detail now and also the phrase "local density changes of each particle" was replaced.

  CM: ll.193-197 In a second step, the particles positions were binned into spatial histograms whereby the unit weight of each particle is based on the product of its transport $T$ and the density changes per day along the part of the trajectory passing through a certain bin. In other words, all particles passing a certain geographical bin along their way were selected and their density changes per day along the

part of the trajectory passing through this bin, weighted by the individual particle's transport, were summed up.

- L. 191, why the sum of all particles has to be multiplied by the length of each particle's trajectories? What's in days? The "length of the trajectory," or the "cumulative sum of transport multiplied by the length of trajectory?"

  AR: The sum of the transport of all particles has to be multiplied by the length of each particle's trajectories because the transport of a particle is included in the sum at each time step and therefore as often as the length of the trajectory in days. We agree that this part was not written clearly and completely rephrased this sentence.

  CM: ll.197-198 Finally, this was divided by the total Agulhas leakage transport as represented by all particles passing through the region.

- Fig. 2: why not include the P-ACT in the bar plot? It would be interesting to see if P-ACT at all other sections.

  AR: We did not include P-ACT in the bar plot as we wanted to focus on the similarities using different Lagrangian tools and hence between A and P. When including P-ACT the attention is more drawn to the differences due to different experiment designs (Fig. R2). This is, however, an expected result and not a unique finding of our study.

[Figure]

Figure R2: (a) Mean (1958-2014) transport across all sections as shown in Fig. 1 for experiment A in red, P in blue and P-ACT in cyan. The transport of all particles not crossing any section is shown as "Lost" and the transport of all particles leaving the region by crossing the release section again is shown as "Meander". (b) Time series of the Agulhas leakage transport for A (red), P (blue) and P-ACT (cyan).

- I like the reasoning for choosing release year as the reference date, including the evidence of fast transition time to 20E, and the strong mixing between 20E and GH line.

  AR: Thank you.

- L. 405: I understand that thermohaline properties are equally important. Are there more references/citations of increasing heat/salt fluxes?

  AR: Yes, Biastoch et al. (2015) and Rouault et al. (2009) also found an increased heat and salt flux into the Atlantic Ocean south of Africa. We included these citations.

  CM: ll.411-414 A substantial increase in total heat and salt fluxes across the Good Hope section over the last decades is the result of a combined effect of changes in transport and water mass properties of Agulhas leakage (Loveday et al., 2015; Biastoch et al., 2015; Rouault et al., 2009).

- L. 418: ",for example," It's nice to state the drawback/advantage of P/A and the future opportunities to use P, but it seems not to be a good idea to conclude this great work. Maybe some reorganization for the last paragraph.

  AR: Thanks for this suggestion, we have reordered this paragraph and included this aspect at an earlier stage.

  CM: ll.417-421 (..) our results show that Parcels can be used for volume transport estimations. Using the 4th-order Runge-Kutta scheme with Parcels results, however, in longer computing times compared to the experiment with Ariane, but the Parcels team is working on an analytical method, increased efficiency and aims for a parallel version (Delandmeter and van Sebille, 2019). This opens up new opportunities for future studies due to the flexibility of Parcels. In a future study, Agulhas leakage could also be estimated with Parcels based on a variety of differently gridded products, e.g. reanalysis products (...)

**References**

Biastoch, A., Durgadoo, J. V., Morrison, A. K., van Sebille, E., Weijer, W., and Griffies, S. M.: Atlantic multi-decadal oscillation covaries with Agulhas leakage, Nature Communications, 6, https://doi.org/10.1038/ncomms10082, 2015.

Delandmeter, P. and van Sebille, E.: The Parcels v2.0 Lagrangian framework: new field interpolation schemes, Geoscientific Model Development, 12, 3571–3584, https://doi.org/10.5194/gmd-12-3571-2019, 2019.

Loveday, B. R., Penven, P., and Reason, C. J.: Southern Annular Mode and westerly-wind-driven changes in Indian-Atlantic exchange mechanisms, Geophysical Research Letters, 42, 4912–4921, https://doi.org/10.1002/2015GL064256, 2015.

Rouault, M., Penven, P., and Pohl, B.: Warming in the Agulhas Current system since the 1980's, Geophysical Research Letters, 36, 2–6, https://doi.org/10.1029/2009GL037987, 2009.

**Response to Anonymous Referee #2**

*Original reviewer's comments are inserted in black, Author Replies (AR) are added in blue, and Changes made to the Manuscript (CM) are finally listed in grey, whereby page and line numbers refer to the fully revised version of the manuscript.*

This manuscript studies the Agulhas Leakage estimates and changes in its thermohaline properties after leaving the Agulhas Current and before entering the South Atlantic. The authors use two offline Lagrangian tools, Parcels and Ariane, based on the velocity field obtained from a 1/20 degree ocean sea ice model covering 1958-2014. They find a robust estimation between the two tools regarding the variability and trend of the leakage, although the mean (climatological) value could vary a lot. They also identified cooling and freshening occurs when the water moves from the Indian Ocean towards the Atlantic Ocean, and a density increase since the thermal effect dominates.

This work confirms the results from Parcels, which is recently developed, are overall consistent with those from the well-established tool Ariane. This encourages the future applications of Parcels as it is getting more and more widely accepted by the community. This works also compares and discusses different experimental designs in the leakage estimation, which is insightful. The presentation is very clear. I would recommend publication of the work only with a few minor suggestions.

AR: Thank you for your positive reply. Your specific comments below pointed us to some ambiguities that we think could be resolved. We want to point out that we do not fully agree with the summary above. It is not only the variability and trend of Agulhas leakage that agree well between the two Lagrangian tools, but also the mean (climatological) transport. The mean transport varies, however, with different designs of the Lagrangian experiment, but this is independent of the Lagrangian tool. We modified the relevant part in the section Discussion to state that more clearly.

CM: ll.386-389 All of these test cases show that changing certain parameters of the Lagrangian experiment can affect in particular the total transport of Agulhas leakage, even if the same Lagrangian tool is being used, and that designing such an experiment to estimate Agulhas leakage is not straight forward due to the turbulent regime.

- L52: The discussion of Lagrangian particles vs. Eulerian tracers is not clear to me. The authors first say both methods are 'widely used', then the authors say the tracking of Lagrangian particles is 'the most widely used'. This is confusing. I understand the authors want to say they are used to estimate different things. Please consider rewriting this part.

  AR: We deleted 'widely' when referring to both the Lagrangian and tracer based approaches. The Lagrangian method is certainly used the most as it the most flexible method. When using an additional passive tracer, a modification of parameters of the experiment would require a rerun of the ocean general circulation model which is more expensive than conducting an offline Lagrangian experiment.

  CM: ll.52-54 As a result, a Lagrangian approach or tracer based estimates in ocean models are used to analyse the variability, trends and characteristics of Agulhas leakage in more detail.
  ll.56-57 The tracking of particles with offline Lagrangian tools is the most widely used approach to estimate Agulhas leakage in ocean models due to its flexibility (e.g. Doglioli et al., 2006; Biastoch et al., 2008; van Sebille et al., 2009).

- L62: Somewhere in the Introduction, it will be nice to explicitly state that both Ariane and Parcels are offline tools instead of online. (Or maybe they can also be implemented into the GCM and run online?)

AR: We have now included that already in the beginning of the paragraph about Lagrangian tools (see changes to lines 56-57 above) as all the mentioned tools (Ariane, CMS and Parcels) are offline tools.

- L113: It is mentioned the hindcast simulation using JRA55-do covers 1958-2019. But here it says 1958-2014. Did I miss anything?

  AR: The JRA55-do forcing does cover the period 1958 to present (currently until mid 2020), but at the time of conducting the hindcast simulation in INALT20 only the period 1958-2018 was available. As particles are advected for 5 years, the last release year is 2014 and as we usually use the release year as the reference date, this results in a time series for Agulhas leakage from 1958 to 2014.

  CM: ll.93-94 Output from a hindcast simulation with the eddy-rich ocean-sea ice model configuration INALT20 (Schwarzkopf et al., 2019) from 1958 to 2018 was used to conduct offline Lagrangian experiments.
  ll.102-103 Here, atmospheric boundary conditions are given by the JRA55-do forcing data set covering the period from 1958 to the present (Tsujino et al., 2018).

- L136: What if a single simulation of 57-years is performed with the particles continuously released at the 32S section? What is the advantage of the current design compared to this one? Could a recirculated particle that could 'pollute' the source be the reason?

  AR: The current design ensures that all particles are advected for exactly the same time period (here 4 years). In addition, the fast transit time through the area and the low number of particles that do not reach any of the boundaries of the area within 4 years (ca. 2%), demonstrate that an advection for 4 years is long enough. A particle that is advected long enough could potentially become part of the Agulhas Current again and cross the release section at 32°S at a time when another particle is just being released due to the continuous release. This would artificially increase the transport at that section as 2 particles represent the same part of the transport and is against the idea of determining Agulhas leakage as a direct connection between the Indian and Atlantic Ocean through the Agulhas Current.

- Fig. 1: It will be nice to label Cape Basin on the map. Plus, please indicate what the red dots represent in the caption.

  AR: We added another panel to Fig. 4 in the manuscript (Fig. R1 in this document) with a map, where the Cape Basin is labelled. We also adjusted the caption of Figure 1.

  CM: In caption to Figure 1: Release positions of particles (shown as red dots) in experiment (b) A and (c) P with a shading of the number of particles per grid cell.

- L411: There are two 'for example's in this sentence. Please consider rewriting.

  AR: Thanks for pointing that out. We have deleted one 'for example'.

  CM: ll.420-422 Agulhas leakage could also be estimated with Parcels based on a variety of differently gridded products, e.g. reanalysis products, which do not have a non-divergent velocity field as required by Ariane.

[Figure]

Figure R1: Comparison of the year of reference used to calculate a time series of the Agulhas leakage transport for experiment P: time series of Agulhas leakage transport referenced to the release year (blue in (a)), year of crossing 20°E (cyan in (a)), year of the first crossing of the Good Hope section (purple in (b)) and year of last crossing of the Good Hope section (green in (a) + (b)). Dashed lines show the combined transport of all particles that cross the Good Hope section an odd number of times, while for the solid lines this additional criterion was not applied. (c) Schematic path of the Agulhas Current via the Retroflection into the Agulhas Return Current (black arrow) and all the sections used for assigning the reference date in (a) and (b). The Cape Basin is located northwestwards of the Agulhas Retroflection and southeastwards of the Good Hope section.

**References**

Biastoch, A., Lutjeharms, J. R., Böning, C. W., and Scheinert, M.: Mesoscale perturbations control inter-ocean exchange south of Africa, Geophysical Research Letters, 35, 2000–2005, https://doi.org/10.1029/2008GL035132, 2008.

Doglioli, A. M., Veneziani, M., Blanke, B., Speich, S., and Griffa, A.: A Lagrangian analysis of the Indian-Atlantic interocean exchange in a regional model, Geophysical Research Letters, 33, 1–5, https://doi.org/10.1029/2006GL026498, 2006.

Schwarzkopf, F. U., Biastoch, A., Böning, C. W., Chanut, J., Durgadoo, J. V., Getzlaff, K., Harlaß, J., Rieck, J. K., Roth, C., Scheinert, M. M., and Schubert, R.: The INALT family - a set of high-resolution nests for the Agulhas Current system within global NEMO ocean/sea-ice configurations, Geoscientific Model Development, 12, 3329–3355, https://doi.org/10.5194/gmd-2018-312, 2019.

Tsujino, H., Urakawa, S., Nakano, H., Small, R. J., Kim, W. M., Yeager, S. G., Danabasoglu, G., Suzuki, T., Bamber, J. L., Bentsen, M., Böning, C. W., Bozec, A., Chassignet, E. P., Curchitser, E., Boeira Dias, F., Durack, P. J., Griffies, S. M., Harada, Y., Ilicak, M., Josey, S. A., Kobayashi, C., Kobayashi, S., Komuro, Y., Large, W. G., Le Sommer, J., Marsland, S. J., Masina, S., Scheinert, M., Tomita, H., Valdivieso, M., and Yamazaki, D.: JRA-55 based surface dataset for driving ocean–sea-ice models (JRA55-do), Ocean Modelling, 130, 79–139, https://doi.org/10.1016/j.ocemod.2018.07.002, 2018.

van Sebille, E., Barron, C. N., Biastoch, A., Van Leeuwen, P. J., Vossepoel, F. C., and De Ruijter, W. P.: Relating Agulhas leakage to the Agulhas Current retroflection location, Ocean Science, 5, 511–521, https://doi.org/10.5194/os-5-511-2009, 2009.